



# The importance of raising risk awareness: lessons learned about risk awareness sessions from the Mediterranean region (North Morocco and West Sardinia, Italy)

Ante Ivčević[1,2], Hubert Mazurek[1], Lionel Siame[3], Raquel Bertoldo[4], Vania Statzu[5], Kamal Agharroud[6],
Isabel Estrela Rego[7], Nibedita Mukherjee[8], Matthieu Kervyn[9], Olivier Bellier[3,10]

[1]Aix Marseille Univ, IRD, LPED, Marseille, France
[2]Priority Actions Programme/Regional Activity Centre (PAP/RAC), UN Environment Programme/Mediterranean Action Plan
(UNEP/MAP), Split, Croatia
[3]Aix Marseille Univ, CNRS, IRD, INRAE, Coll France, CEREGE, Aix-en-Provence, France
[4]Aix Marseille Univ, LPS, Aix-en-Provence, France
[5]Mediterranean Sea and Coast Foundation (MEDSEA), Cagliari, Italy
[6]Département de Géologie, Faculté des Sciences de Tétouan, Université Abdelmalek Essaâdi, Tétouan, Morocco
[7]Research Institute for Volcanology and Risk Assessment (IVAR), University of the Azores, Azores, Portugal
[8]Brunel University London, United Kingdom
[9]Department of Geography, Vrije Universiteit Brussel, Brussels, Belgium
[10]Aix Marseille Univ, CNRS, ECCOREV, Aix-en-Provence, France

*Correspondence to*: Ante Ivčević (Aix-Marseille Université, St Charles, case 10, 3 place Victor Hugo, 13331 Marseille Cedex
3, France; email : ante.ivcevic@univ-amu.fr)

**Abstract.** Risk management is used in societies to mitigate the potentially dramatic effects of natural hazards. Local authorities
and managers use different indicators in elaborating rescue and urbanism plans, which are not always efficient in reducing
impact in the time of the crisis. This highlights society's vulnerability in the particular context of global environmental and
climate changes. This interdisciplinary research aimed at identifying reliable risk indicators and societal responses regarding
natural hazards and climate change impacts, to provide a governance framework for disaster risk reduction. Different societies
face diverse risks and do not necessarily have the same level of local awareness confronting them. Two sites were thus selected
from the Mediterranean basin, one from the South coast (North Morocco), other from the North coast (the Italian island of
Sardinia). North Morocco, the region of multi-risks, is characterized by high demographic and economic pressures; West
Sardinia counts for remarkable biodiversity of wetlands and is characterized by high environmental and agricultural pressures,
which in both cases intensify the vulnerability of the coastal areas. Testing for the local population's preparedness for future
financial protection showed the importance of risk awareness sessions as an indicator of risk management. The significance of
risk awareness sessions is demonstrated in a quantitative part of the study with the local population, and their importance is
also discussed with other stakeholders in North Morocco in a qualitative part of the study. Based on these findings, further
ideas on a new series of less descriptive, more dynamic, and more user-friendly indicators are suggested. How can risk sessions
be a dynamic indicator of a resilient society? The obtained results could serve in future governance frameworks for the
mitigation of natural hazards in the Mediterranean region and wider. Finally, the urgent need for continuous work to overcome
the communication gap between the scientific community, risk administrators, civil society and the general population is
encouraged.

# 1 Introduction

Different societies sharing a similar environment may face similar natural hazards, but might not necessarily have the same
level of risk awareness. They might not share a common understanding of that risk. Actually, vulnerability of societies is a



dynamic process, always relative to a certain situation and depends on local knowledge, practices, and cultural specificities (Moscovici, 1984). The importance of local knowledge for sustainable disaster risk reduction and management is largely recognised by international policy, such as the Sendai Framework for Disaster Risk Reduction (UN, 2015). However, the extent to which this local knowledge is translated into concrete, effective and practical measures is still open to debate (Fischer, 2000; Bwambale, 2020).

More than 500 million people are estimated to live currently in the countries bordering the Mediterranean Sea (Karadirek et al., 2019), with around 150 million people living along the Mediterranean coasts (Cramer et al., 2018). The Mediterranean basin is one of the cradles of civilisation and has always been an attractive migration hub, nowadays also the most visited tourist destination in the world, with more than 330 million visitors in 2016 (Tovar-Sánchez et al., 2019). This region is prone to many climate and non-climate induced hazards (Satta et al., 2016), and due to a very dense population and infrastructure, a

vulnerability hotspot. It is one of the regions most vulnerable to global warming, with a forecasted temperature rise of 2-3°C by 2050 (Plan Bleu, 2019). The region is projected to experience the highest increase in the frequency of occurrence of extreme weather events due to climate and environmental change (Vousdoukas et al., 2017), and the open question is whether the population is aware of those imminent risks.

In this paper, North Morocco and West Sardinia, two case studies from the Mediterranean basin, are compared, in order to

explore what could contribute to increased precautionary behaviour in a form of future financial readiness to protect of the local, general population. These two studies provide the opportunity to compare how participating in risk awareness sessions is translated into practice, where risk awareness sessions stand for educational training or workshops of the local population regarding local natural hazards. A face-to-face questionnaire-based survey of the general population is followed by interviews with the administrators, scientists and associations and the policy literature consultation, to confirm or reject the importance

of organizing and assisting to risk awareness sessions in the actual risk management in the regions. This is important to predict the advantages and flaws of the present risk policy and to overcome a possible risk management gap. Analysis of these two field studies inform us about the Mediterranean region and sustain a deeper reflection about how these common risks could contribute to risk management strategy in other interregional locations. Feasible policy advice related to multi-risks and risks related to climate change is formulated as a conclusion, which could contribute to disaster risk reduction in light of the UN

Sendai's recommendation, preventing future human and financial loss.

## 1.1 Risk awareness

Risk perception plays a leading role in efficient risk management. In one of the first contributions of cognitive psychology to decision making under risk, the concept of risk perception depicted that non-expert people might perceive risks by resorting to different logics than experts do (Slovic et al., 1976), making the process of risk appropriation more complex. Likewise,

Covello (1983) suggested that a better understanding of how experts and non-experts reflect on and decide about risks would be beneficial to both risk analysts and decision-makers. In the later article 'Perception of risk', Slovic (1987) proposed a strategy for studying perceived risk through a psychometric paradigm, which uses multivariate analysis techniques and scaling



to produce a quantitative representation of risk perceptions. The main conclusion is that laypeople can be short of some information related to hazards, but have a rich perception of risk that has to be included in risk assessments by the experts.

Rohrmann (1998) defined risk perception as a concept that "refers to people's judgments and evaluations of hazards they are or might be exposed to" which are the "interpretations of the world, based on experiences and/or beliefs". Since the images of risk are sometimes distorted, public perception is at least partly driven by biases, false assumptions and sensation, as argued by Renn (1998). He reasoned that risk perception studies help to collect different personal experiences with risks and to identify public concerns. However, they cannot state any normative legitimacy. Since scientific expertise and rational decision-making

are necessary, but not sufficient, such a debate between experts and non-experts is needed.

Another discipline that has a say in risk perception is applied social psychology. As argued in the paper by Langford (2002), social and cultural settings need to be studied to determine how anxieties relate to the ways people perceive risks. Among his case studies, one related to climate change captured a wide range of personal, social and environmental anxieties. This helped him to argue that the complexity and importance that risk perception has in forming identities demand people's recognition

and involvement. People need to be persuaded in their need to act, and only by believing in change can risk managers develop significant and far-reaching risk strategies.

When determining the indicators in risk management, the awareness, perception and behaviour need to be distinguished. Although sometimes the concepts of risk awareness and risk perception are overlapping or even considered as synonyms, here the concepts draw from the distinction made by Papagiannaki et al. (2019a, 2019b). Risk awareness is, therefore, reflected as

the measure of information and knowledge uptake. Risk perception is related to conceptual understanding of the threat, based on a personal interpretation of the knowledge arising from a cultural background, and risk precautionary behaviour is the result of awareness and perception based on the socio-economic context. A recent study on hydrogeological risk awareness and preparedness in Italy (Mondiano et al., 2020) concluded that lack of big events and poor risk communication strategies can cause risk awareness to decrease. This shows that risk awareness is essential for adequate risk management policies and needs

to be further explored. In addition to risk awareness, the cultural context will be tested by comparing two different societies, from different socio-economic contexts, inspired by a cultural approach that considers risk as a social and cultural construct (Bertoldo, 2021). Cross-cultural differences between North Morocco and Sardinia will be detected, and the actual risk behaviour of North Moroccans and Sardinians will be described, in line with Rohrmann (1998). Since it was shown long ago that the communication and the transfer of information contribute to different risk perceptions between the experts and non-

experts (Slovic et al.,1976, Covello, 1983, Renn, 1998), the efficiency of risk communication between the general population, scientists and managers will also be questioned.

## 1.2 The study sites

The selected case studies are North Morocco and Sardinia in the Mediterranean basin (Fig. 1). Being the region prone to many natural hazards and a climate change 'hotspot', the concern grows with the increase of population in already densely populated

Mediterranean basin.



The region of North Morocco is a multi-risks zone, which is worth studying due to the increased complexity of social and natural processes. The region is prone to natural hazards particularly due to the seismic activity in the Gibraltar/Alboran region, between the Nubian and Iberian plates, that hosted disastrous earthquakes like 1755 Lisbon and 2004 Al Hoceima events (Neres et al., 2016). In addition, due to natural predispositions (lithology and slopes) and land use of the Rif Mountains, as well as the location on the transition between different climate types, the area frequently suffers from landslides, erosion, droughts, floods and flash floods (Ivčević et al., 2020a). The region's coasts are also of decreased resilience to climate change due to the heavy urbanisation of the coastlines (Snoussi et al., 2010), that are highly exposed to sea-level rise (Satta et al., 2016) and possibly tsunamis (Benchekroun et al., 2015). During the last decades, the vulnerability of the region increased since the region became a big economic hub at the end of the 20th century, with a touristic infrastructure boom (Fig. 2) and the project of harbour Tanger-Med. Nevertheless, Morocco maintains, in its modernity, very strong traditional components, with the traditional forms of land use (El Abdellaoui et al., 2008). The population lives of land, and the agricultural sector is the backbone of the regional economy (Salhi et al., 2020). Besides, Morocco is a Muslim society, with religion playing a significant influence on opinions and perceptions of risk, particularly seismic one (Paradise, 2008).

The coasts of West Sardinia are amongst one-third of areas exposed to the highest level of hazard in the Mediterranean basin, mostly due to very high wave heights (Satta et al., 2017). The area of the Gulf of Oristano is frequently hit by extreme weather events, above all flash floods and coastal storms, predominantly during the cold season. Also, a maximum sea level increase of 949 mm relative to current sea level is projected in 2100 for the Gulf of Oristano, based on the IPCC AR5 8.5 scenario, which could leave some parts of the gulf partially flooded (Antonioli et al., 2017). Regarding flooding, wetlands in this area (Fig. 3) play a vital role in disaster risk reduction, but they also create a conflict between urban, fishing and farming activities, with the town of Cabras famous for fishing, and the fertile area of Arborea, home to an extensive agricultural production. In addition, Sardinians have a long tradition of autonomous governance and environmental awareness, which even included wildfire regulations in the 14[th] century (Ivčević et al., 2020b).

## 1.3 Summary and goals

Each territory is unique, and risks are socially constructed and locally managed. Local knowledge is recognised as valuable in the efficiency of disaster risk reduction of resilient societies. This is true at least theoretically, as promoted by the Sendai Framework for Disaster Risk Reduction (UN, 2015) and other international policies dealing with climate adaptation (Trogrlić et al., 2021). Accounting for local risk awareness contributes to tailoring risk management policies compatible with the core values of the communities at stake, hence facilitating the support of the local population. The main questions we, therefore, ask in this paper are: Are locally organised sessions dedicated to increasing risk awareness a predictor of societal resilience facing risks? More precisely, are the participants in risk sessions associated with higher risk awareness and precautionary behaviour?

This comparative quantitative and qualitative study contributes to explaining risk preparedness by comparing the role of risk awareness sessions to all stakeholders included in risk management processes. It describes relations between risk awareness





and precautionary behaviour in two areas from the Mediterranean basin, North Morocco and West Sardinia. It also explores
the role of risk awareness sessions in explaining individual future monetary behaviour. Finally, it discusses the additional
benefits of risk awareness sessions in the construction of dynamic indicators of risk management.

## 2 Methodology

### 2.1 Participants and fieldwork procedure

The fieldwork procedure was both quantitative and qualitative, organized around questionnaires and interviews. Firstly, to
collect insight on the usefulness of risk awareness sessions with the general population, a questionnaire survey and interviews
were used for the Moroccan case study. The convenience sampling was considered well-suited for the exploratory purpose of
this pilot study regarding local risk awareness. The questionnaire for the general public resulted in responses of 391 inhabitants
of North Morocco from eight different communes. The questionnaires were collected face-to-face with the help of local
enumerators (students) in a public setting, during November 2018. The chosen municipalities belong to areas with a different
historical hazards record and different geographical characteristics and are divided into three areas for the purpose of this
study: Atlantic coast, valley of river Martil and Rif Mountain zones (Fig. 1). The obtained sample is composed of 44.2%
female respondents and thus underestimates the female population (which represents 49.27% of the regional population, census
2014). The respondents are on average between 36 and 41 years old (which overestimates the fact that Morocco is a young
nation with a median age of 29.5 years, census 2014).
In the Sardinian case study, the questionnaire was used for data collection, and was later crossed with the existing secondary
literature on risk management on a national and regional level. Hundred and seventy-six inhabitants of the area of West
Sardinia from three different municipalities, urban Oristano and rural Cabras and Arborea, responded to our questionnaire in
a public setting. This questionnaire-based survey was conducted during May and June 2019. Data were collected face-to-face,
by randomly approaching possible participants, and about 20 min were needed to complete each survey. The Sardinian sample
is composed of 45% of male respondents (figures in the population are 49% from the 2011 census, so our sample
underestimates the male population). The average participant is 46.4 years old (44 is the average age in population, census
2011).
Secondly, the use of interviews in the North Moroccan study was two-fold. They served to introduce the problem of risk
management in the region and to construct the questionnaire. Besides, they are an additional source of information to be crossed
with the responses from the questionnaires. There were twenty-five interviews conducted, and the interviewees were grouped
into three profiles: a) the elected or appointed administrators (eight), b) the scientists and technicians that work in the risk
research or management (eight), and c) the members of civil society, i.e. associations (nine). During the interviews, the
indicators, risk management policy and personal memory of risks in the region of the study were discussed. As far as the
Sardinian case study is concerned, the secondary literature on risk management at the regional (Sardinia) and national (Italy)
level was obtained as public documents free from the administrative web pages. These sources served to contrast the





population's responses with the official directions in managing natural risks. Since this literature was easily accessible, the qualitative study in form of interviews with the local authorities was not considered to be essential for this research, conversely to the Moroccan case study.

## 2.2 Quantitative part of the study: Survey content and variables

The Moroccan and Sardinian questionnaires consisted of 39 and 30 questions, respectively, and were divided into five sections. Not all questions from the questionnaires were used in this analysis. The common part of the questionnaire were the sections that dealt with risk awareness, personal experience, formal and informal education on risks; in addition to questions regarding institutional trust, environmental identity and place attachment; and finally questions about the belief in climate change, adopting precautionary measures and investing money. Those questions were close-ended (dichotomous and Likert rating

scale). The full questionnaires (in French and Italian) are available upon request. The data were described in Excel and SPSS, the latter used for correlations and regression model. Since the data do not follow a normal distribution, correlations were verified using Spearman's test. The dependent variable in the regression model was the future readiness to protect, more precisely the yes/no question if the respondents were truly ready to invest money to protect themselves in the future. The used model was binary logistic which allows dividing variables into more blocks, tracking the model's progress. All the variables

were nominal (Table 1).

  The first set of variables used in the regression model consists of demographic ones: gender (value 1 for women), age (value 1 for those respondents older than mean age of 36 years old), education (value 1 for those with bachelor's degree or higher), and zone of living (Morocco: Atlantic, Martil, Rif, Sardinia: urban and rural, five dummies for five zones). The second block includes explanatory variables related to risk awareness that are expected to inform future readiness to invest money, as an

indicator of future monetary precautionary behaviour. Risk awareness was described with two indicators: one related to the level of information about natural hazards (each hazard listed and evaluated on the Likert 1-5 scale), other related to risk sessions (whether the respondents participated or not in the awareness sessions or environmental education campaigns organized by the municipalities or associations), (e.g., Domingues et al., 2018; Ivčević et al., 2020a). Besides, the variable of personal experience of a natural hazard (yes/no) was added, whose importance in risk perception was showed in similar studies

(e.g., Papagiannaki et al., 2019b). Moreover, respondent's belief in the existence of climate change was asked (yes/no). Furthermore, the respondents were asked if they have previously taken any precautionary measures (yes/no). The final block of explanatory variables, related to trust, environmental identity and place attachment, was to test the influence of these supplementary variables since they might play a role in future precautionary behaviour. The place attachment was measured using items such as 'I think this municipality offers a good living environment' and represents a relationship of resident with

her municipality (Bonaiuto et al., 2016). The environmental identity was measured using items such as 'I think of myself as someone engaged in environmental issues' and represents environmental concerns of residents (Bertoldo and Castro, 2016). The social trust, regarding science and institutions involved in risk management, is of importance especially when respondents





cannot manage risks on their own (Achterberg et al., 2017). All those items were measured using a 5-point Likert scale, and then dichotomized: participants with responses above mean were considered to have a high value, those under mean to have low value, and are as such used in a binary logistic model. For more details on variables chosen refer to publications by Ivčević et al. (2020a, 2020b).

## 2.3 Qualitative part of the study: Interviews and secondary literature

In North Morocco, the interviews were based on an interview guide. The general topic was focused on the influence of the representation that members of associations, administrators and scientists have on the relationship between citizens and risk; and on the influence of memory and the territorial context in the development of management tools, particularly preventive communication. The administrators were also asked about the indicators used in risk management. Besides the topic of risk experience and risk memory, the interlocutors were asked about the different types of knowledge (scientific and vernacular) that the institutional practices base their risk awareness activities on when talking about risks to the general population. Also, the importance of risk awareness sessions in risk management strategies was discussed. All twenty-five narratives were recorded and transcribed (Ivčević et al., 2020c).

## 3 Results

### 3.1 Quantitative: Explaining future willingness to invest money based on actual risk awareness

Table 2 presents the correlations of five different areas (binary variable for each area) of the study with main indicators. The zones Martil and the Rif are positively correlated with 'personal experience', unlike Sardinian areas that are negatively correlated with the same variable. Sardinian areas are also negatively correlated, and the Rif positively correlated with 'risk sessions'. Martil area is positively, and urban Sardinia negatively correlated with the variable 'trust science'. All three Moroccan zones are negatively correlated with the variable 'trust state', conversely to Sardinian zones that are positively correlated with the same variable. Rural Sardinia is positively correlated with both 'place attachment' and 'environmental identity', whereas the Rif is negatively correlated with the former and Martil with the latter variable. Martil area is also negatively correlated with variable 'invest money', similarly to the Atlantic area with the variable of 'precautions taken' (Table 2).

Readiness to invest money in order to protect from future natural hazards was examined through a binary logistic model. Readiness was regressed on three blocks of variables. The first block included demographic variables of gender, age, education and zone of study. The second block included risk awareness variables that were considered to contribute more to explaining future willingness to invest: personal experience, precautions taken, risk informed and risk sessions. The last block included additional variables that hopefully could help clarify which profiles are up to risk investing: trust in science and state,





environmental identity, place attachment, and belief in climate change. Presented in Table 3, results show that demographic

variables that significantly predict future readiness to invest money in protection are the age of the respondent over 36 years

old *(-.433, p < .05)*, and the education level of any university degree *(.459, p < .05)*. This result suggests that being under 36

years old and having a university education is positively associated with monetary investments for natural risk protection. The

only additional indicator that predicts future willingness to invest is the awareness indicator of following risk sessions *(.539,*

*p < .05)*. Personal experience is a marginally significant positive predictor *(.396, p < .1)*, as well as the fact of living in West

Sardinia – in both urban (*.780, p < .1*) or rural areas (*.743, p < .1*). None of the indicators introduced in the third block did

significantly contribute to future willingness to invest money. The model overall describes over 11% of the social variability

of the sampled population (Nagelkerke $R^2$ = .115).

### 3.2 Qualitative: The importance of the risk awareness sessions according to the stakeholders

The interviews served not only to learn about the region and to build bridges with local contacts in the region of North Morocco,

but also to contrast the general population's responses with the official directions in managing natural risks. The interlocutors

have been asked about natural hazards, their memory of catastrophic events and the risk management policy in the region.

Among the useful information given, some particular answers need to be discussed. During these interviews, the main topic

elaborated were the indicators of risk management. As far as risk indicators are concerned, the administrators from smaller

municipalities do not consider themselves in charge of risk management, possibly due to centralization of power, which is

either in the region's capital Tangier or in the national capital Rabat. Male Administrator (55 y.o. from Oued Laou) confirmed:

*"Wilaya produces the ORSEC and therefore I have no idea about the indicators."* ORSEC[1] is an emergency response in case

of disasters, here related to floods. Besides, the concentration of power is demonstrated in the term '*wilaya*' which is an

administrative unit, headed by a *wali*, appointed by the king. In short, the higher entity '*wilaya*' is in charge of risk

management, and the small municipality is not informed about it. A similar signal comes from the municipality of M'Diq:

*"The indicators are not used [here], there is a specialized service that does this."* (Male Administrator, 60 y.o.).

During interviews, scientists, however, called for more investment and more studies, but also underlined what they do not

have: *"[...] politicians want impact indicators and we cannot yet provide them since it is expensive and because we lack the*

*expertise. For example, recently we have raised awareness in 40 schools, but after all, we do not know the success rate of*

*schooling."* (Male Scientist, 50 y.o., Tetouan). The same scientist continued on awareness: *"The problem is that people forget,*

*we need more awareness [campaigns], we need to train people at the institutional level. Also, everything that worked in theory*

*during the disaster on the spot falls ... So the risk culture must be developed."* His colleague confirmed that people are aware,

but they choose hazardous behaviour: *"People are of course very aware of the risks [...] but when it comes to economic*

*interests, everyone forgets about the risks. The [Martil] valley floods are very well-known among the population, everyone*

*knows and yet everyone forgets it when it comes to economic interest."* He continued about the awareness campaigns:

---

[1] Organisation de la Réponse de SEcurité Civile = Organisation of the Civil Security Response



*"Unregulated awareness-raising among the population (schools, associations) is useless, it should be in the law. To do an*
*intervention you need authorisation and ABHL[2] was already refused to avoid controversy. In 2016, we participated with the*
*wilaya on awareness-raising but it was mediocre. Officially in education, there is no notion of risks, only a few individual*
*initiatives done voluntarily."* (Male Scientist, 30, Tetouan).

In the region, the 2004 Al Hoceima earthquake is still fresh in the memory of the population: *"After the 2004 disaster, half of*
*the population sought help from a psychologist. I myself stayed with my family for 1 month in a tent."* (Female Administrator,
60 y.o., Al Hoceima). The administration is aware of the need for raising risk awareness: *"The population is aware of this*
*problem, in any case, the municipalities are obliged according to the law, and municipal charter, and even according to the*
*constitution to raise awareness of the population for the prevention of various risks."* (Male Administrator, 40 y.o., Martil).
This leaves some hope, but there is a need to raise awareness in small municipalities and outside of big urban centres as well,
because the awareness campaigns, if they exist, evidently do not arrive in all parts of the region: *"At school, there is no*
*awareness. Even at the mosque, which is a showcase, in the speech of Friday, never a speech on the risks was mentioned. So*
*there's nothing, total absence."* (Male Administrator, 55 y.o., Oued Laou)

However, regarding risk awareness, the civil society expects more from the administration: *"The city does not organise the*
*awareness-raising sessions, but there are nature protection associations and they organise the seminars."* (Female Associate,
50 y.o., Tetouan); *"Even the administration does not raise awareness; they arrive after the event to remedy it, but not before*
*to anticipate it. The administration must have the action and emergency plan."* (Male Associate, 55 y.o., Chefchaouen).

As far as the Sardinian case study is concerned, risk awareness sessions among the local society experiencing and managing
risks seem to be taken into consideration in the official documents, but in a form of a theoretical wish list. In 'Elements for
National Strategy of Adaptation to Climate Change' (Castellari et al., 2014), the authors recognised the importance of risk
perception: "investments in innovative monitoring technologies, investments to improve communication and to raise
awareness of perception and risk management by citizens (an informed population is more aware and safer), they are priority
measures of 'non-structural' adaptation in the context of climate change" (p. 38) and "an effective communication, awareness
and information activity to the interested parties for the purpose is, therefore, essential to ensure an adequate perception of
these risks" (p. 216).
Similarly, in the Sardinian 'Regional Strategy of Adaptation to Climate Change' (adopted in 2019), among the general criteria
in developing the objectives is the "awareness and education on climate change, to ensure full awareness of the future risks
associated with expected climate pressures and to stimulate responses also oriented towards the development of bottom-up
planning tailored to specific local needs" (p. 37). The strategy identified sets of indicators to describe climate changes and
their consequences on natural and human-made systems. The used indicators deemed relevant for the study of changes in the
characteristics (frequency and intensity) of particular impacts must be followed up for the climatic analysis. This document is

---

[2] Agence du Bassin Hydraulique du Loukkos = Loukkos River Basin Agency



followed by an extensive and detailed 'Methods and Tools for the Regional Strategy of Adaptation to Climate Change' (2018), where indicators useful for knowledge of the state of the environment in Sardinia are listed, among them the farmers' perception of the threats related to climate change is elaborated. In addition, a document 'Guidelines for Regional Adaptation Strategies', elaborates the role of the coordination group, and dedicates a place for questionnaires and workshops in the process

of adaptation: "developing specific surveys with questionnaires or semi-structured interviews, focus groups, participatory workshops, laboratories managed with methodologies relating to participatory planning, etc" (Cocco et al., 2019). Finally, a document 'Guidelines, Standardized Principles and Procedures for Climate Analysis and the Assessment of Vulnerability at Regional Level', from the same MASTER ADAPT project (Giordano et al., 2019), calls for the quality of input data and states that "[a] deeper understanding of how a sector / system / territory behaves with respect to climate changes contributes, in fact,

to establishing adaptation objectives and targets, to providing the necessary elements for planning adaptation measures, to raising community awareness and to monitoring and evaluating adaptation policies" (p. 31).

## 4 Discussion

### 4.1 Regional contrast in risk awareness

The correlations between each of the five different areas of this study and the main chosen indicators capture the signal of regional differences between our two case studies within the Mediterranean basin. This regional variability indicates a different level of risk awareness between the two case studies: respondents in Morocco have a stronger risk memory while those in Sardinia are more informed and educated about risk. Respondents from both urban *(34%, -.087, p < .05)* and rural *(29%, -.146, p < .01)* Sardinian areas report less, whereas respondents from Moroccan Rif area report more generally to have attended

risk awareness sessions *(57%, .220, p < .01)*, a result that demonstrates a similar dynamics as 'personal experience'. It is rather surprising that correlations suggest risk awareness sessions being more common in Morocco than in Sardinia, which is contrary to qualitative data presented. This contrast could be explained by the difference in hazard history that those areas experienced, although this is not obvious. Inhabitants of the Rif have in memory a recent series of earthquakes, with a notorious and deadly event in 2004. Conversely, Sardinians seem to be less impacted by what they experienced and learned about regarding their

territory. These differences of risk experience may lead to differences of main hazards perceived between two studied regions (Fig. 4).

### 4.2 All stakeholders underline the importance of risk awareness sessions, but does it remain only a wishful thinking?

The results from quantitative case studies in North Morocco and Sardinia, targeting the general population, indicate that participation in risk sessions seem to have an impact on precautionary behaviour, related to future readiness to investing money

in order to protect *(.539, p < .05*, Table 3). However, it might be that people more interested or scared by risks are more likely





to attend training session and are more willing to invest in precautionary measure. The importance of risk awareness sessions is also confirmed by interviews in the qualitative study in North Morocco. Although both scientists and members of associations call for more awareness campaigns, the majority of administrators defend, however, the position that the population is already aware of risks and that the institutions are well-involved. Still, an administrator from a smaller

municipality admitted the absence of sessions and learning processes on risk. The risk awareness sessions in risk management are elaborated by all three groups of interlocutors. As stated by the administrators in Rabat, Martil or Al Hoceima, the risk sessions exist and the population is aware of the risks, but in Oued Laou or Tetouan they do not agree, where the latter stated *"[...]this question of risk is not a regulatory, institutional or scientific problem, it is a problem of raising public awareness. The population is not susceptible, they want to build."* Similarly, the scientists talked about partial awareness, where one of

them from Al Hoceima underlined that *"awareness exists, but there is still a high illiteracy rate so we are making a great effort to explain it to everyone."*

On the national level, the Moroccan state is well-included in risk management, as confirmed by an administrator from Rabat: *"Morocco has positioned itself as the hard-working student for the fight against climate change and the protection of the*

*environment within the COP21. The State is well involved, the blue pavilions, the institutions, ... with specific awareness training, workshops organized, [...]. So the awareness is there and it should not be overlooked, in addition with the installation of the Fund to fight against natural risks."*

Besides, the Moroccan 'Référentiel de l'Urbanisme Durable'[3] (2017) elaborated the detailed commitment for which Morocco engaged by adopting the new urban agenda 'Quito Habitat III' from October 2016. The fifth issue, dedicated to 'Health and

Citizens' Security', includes a special theme on 'Management of Natural and Technological Risks', with natural hazards like avalanches, wildfires, floods, landslides, cyclones, storms, earthquakes and volcanic eruptions. The listed objectives of prevention are: to suppress the extent of the hazard, to control its effects and to reduce the exposure of people and goods. The strategy of implementation is rather ambitious, with action, intervention, urgency and rescue plans, followed by the financial investments, to implement alerting systems, to develop risk cartography, to conform the constructions following the

earthquake-resistant standards, to integrate risk management into local action plans and to realise infrastructure resilient to risks.

However, there is some void that is argued by the interlocutors, starting from the lack of legal respect, to lack of awareness and trust. Firstly, although the State's implication on a national level was appraised by the administrators, laws are there, but they are not respected, as stated by the scientists from Tangier (*"At the level of the law everything exists, the problem is if we*

*respect the law or not. The regulations exist, but they are not followed strictly enough."*) and Martil (*"[...] the population [...] they do not respect the law. You have to make people respect the law."*). The population seems to disobey the legislation, as well as the local leaders, when it comes to financial matters and corruption, which creates distrust, as elaborated by the scientist

---

[3] Sustainable Urbanism Reference System



from Tetouan (*"The land was expropriated for non-construction, but now according to the development plan it is no longer. The state still decided to build hotel and tourism projects. People are of course very aware of the risks [...] but when it comes* 
*to economic interests, everyone forgets about the risks…"*). Is the local population aware of risks or just decides based on an economic opportunity? Based on another scientist from Tetouan, it could be considered that awareness and financial means are actually intertwined: *"In construction, the responsibility lies with the architect. But not everyone passes by the architect and not everyone meets the standards, even if the architect is there. On the other hand, yes, that poses problems. But this is not due to the wrongdoing, but due to the mentality of the inhabitant himself. I don't think there's awareness among the locals.* 
*What we see is that customers want the more space possible."* The financial resources are also likely to influence the risk management that is deficient at the municipal level. As stated by smaller municipalities of Oued Laou and M'Diq, they do not use indicators of risk management therefore they do not manage risks on their own. In addition, the lack of resources, financial but also human, is underlined by the administrator from Oued Laou: *"We are 4 civil servants for 9665 inhabitants (2014) and during summer [this number] is tripled."*

Secondly, a remarkable obstacle in effective risk awareness campaign is the reciprocal distrust between administrators and scientists. Previously mentioned distrust of local leaders towards awareness session proposed by scientist from Tetouan's ABHL that was refused authorisation to avoid controversy, is confirmed by the researcher working in the national research institute CNRST[4] in Rabat. While conducting her research on tsunami hazard in Tangier *"I retained by working with the managers in Tangier on the tsunami simulation is that initially by talking with them they are very understandable and they* 
*will follow up to the goal, that it is enough to make a request, and when D-Day comes they are reluctant and afraid to speak. Managers are afraid to scare tourists and the city's resident population. Morocco's economy is based on agriculture and tourism and Tangier is very touristic (4th on the national level). They are afraid that if the tourists look at the tsunami signs in the city that they will think that it is eminent that a tsunami is coming and therefore why bother with the tsunami?"* 
The distrust between the administration and the broad population is mutual. As a female member of association from Tetouan 
stated: *"For management, the most important is to do a good job, to work. It's not to repeat with each election 'we'll do this, we'll do that', and after 4 years the results will be nothing and the loop will come again 'we'll do this, we'll do that'; it is not the continuation of the projects as it is elsewhere, they do not do their job".* A female administrator from Al Hoceima confirmed: *"We work secretly because the population is not interested in public work related to risk, it is the task of the municipality. We did not even communicate the risks to the inhabitants of the cliff."*

To sum up, a point stated by a male member of association from Al Hoceima can be used to depict the situation in the region: *"Natural hazards, yes, people are aware and informed that this region is seismic. Much the same as with garbage, people know that it's not good. But what do we do? Is this awareness sufficient to change behaviour, attitudes?"*

---

[4] Centre National pour la Recherche Scientifique et Technique = National Centre for Scientific and Technical Research





As far as Sardinian strategic documents are concerned, all official documents look professional and well-elaborated,
nevertheless without concrete measures of a broad assessment of risk awareness. Based on the results from the quantitative
case study, the indicator of risk awareness sessions seems to be significant in deciding whether or not the local population
invests in future protection. Besides, the lack of risk awareness sessions was often brought out by the respondents on the
margins of the quantitative study. However, actual actions might in fact exist, but the impression is that these strategic
documents do recognize the need for awareness and education sessions, yet without specific training activities. It is as if there
is no follow-up of these wishes in concrete monitoring, which therefore does not provide a useful indicator of climate change
adaptation. This calls for the administration to promote and specify more risk awareness sessions among the general population.

### 4.3 Future perspectives of the results as an input for risk policy: the importance of dynamic indicators

Another risk management recommendation of this study regards dynamic indicators or actions, where risk awareness sessions
particularly contribute. In effective risk management, indicators should be used to collect information on every phase of the
disaster management cycle, building a less vulnerable society. For example, in economics, cost-benefit analysis of the actual
and new risk policy, before and after the investment, could be coupled. Also in education, for instance, the number of middle-
or high-school pupils educated on risk and climate change could be coupled with the number of aware pupils in an affected
area and those that consequently developed a good understanding of climate change processes, which showed to have an
impact on the Sardinian population in adopting precautionary behaviour. Raising awareness in schools was already tested in
Tetouan, as brought out by our interlocutor, but so far without an estimate of the success rate (Male Scientist, 50 y.o., Tetouan).
The information on hazard history, recurrence time of earthquakes or return interval of floods can be used as input data for
risk awareness sessions, that would boost both awareness and perception, and provoke changes in attitudes towards
precautionary behaviour, as was shown in North Morocco. Risk awareness sessions can further contribute to the
interdisciplinarity of risk science with policy, psychology, and economics.
Furthermore, the North Moroccan example showed the need for integrating risk awareness sessions with religious institutions.
The hazard history of great earthquakes and floods could be used as input risk information during religious discussions with
an outcome of an ethical maxim of adopting precautionary behaviour to save human lives. Moreover, the participatory
approach is always important for a successful information flux between all stakeholders included in risk management
processes. It can also be assessed dynamically, by initially identifying the individual positions of the risk management
stakeholders, and consequently reaching consensual priorities of all stakeholders after participatory focus groups.
This way of approaching indicators is a suitable consideration for different hazard types. If we consider flood hazard, and the
number of people living in the area exposed to flood, that would be an example of exposure, or more broadly speaking,
vulnerability indicator. Since it could be assumed that we have the information about flood frequency and maximum or average
intensity, the educational campaign in that area could be organized. The number of people that moved out from that flooded
area based on the input of the educational campaign, in some limited time frame, could be measured and be considered as a
resilience indicator. These pieces of information could notify about the success of the flood awareness campaign, but also



about the local readiness to cope with the ongoing change. This, however, will remain only wishful thinking until a strong policy and compensation scheme supporting the relocation take place, and raising awareness will help only to increase the acceptance of such policy.

Last but not least, if we consider the sea-level rise and coastal storm effects as a consequence of climate change, the community can be informed about the annual cost of beach nourishment and possible control structures. If the coastal storm damage is registered, and the benefits of nature-based solutions to mitigate the impact of climate change divulged to the population, the community could see the utility and financial benefit of nature-based solutions, as it is promoted with the wetlands from the Sardinian case study. There again risk awareness sessions have an important position in the learning process. A similar idea

can be used for the example of geomorphological hazards. These ideas are presented in Table 4 and are some of the possible changes that dynamic indicators could bring to the community.

### 4.4 Limits of this study

The added value of analysing two Mediterranean case studies is the region's North-South coupling and comparison, where Morocco and Italy could easily serve as pilot studies to other countries, sharing the methodology and adapting to local physical

and social particularities. Likewise, undertaking risk perception case studies in lower-income countries help to get out from the mindset of the higher-income country, since financial obstacle can play a role in different forms of precautionary behaviour. The variety of case studies should increase the efficiency of risk management strategies. Still, comparing two areas of study is not always a simple task: risks are complex phenomena and those acceptable in one cultural context are not necessarily acceptable in another (Dauphiné, 2003). In this case, the first to keep in mind while trying to compare these two Mediterranean

territories was their size proportion. North Morocco is a region of more than 17,000 km² and with more than 3,5 million inhabitants. The Sardinian surface is a bit over 24,000 km² with 1,6 million inhabitants. However, the Gulf of Oristano municipalities included in our case study cover an area seventeen times smaller than that of the Moroccan case study, and with forty times fewer dwellers (1000 km², 85,000 people). This is, therefore, the biggest limit of this comparative study. To continue, additional comparative studies from the Mediterranean basin should be organised, from both North and South coasts.

This approach would generate new knowledge that has to be constantly shared in form of risk awareness sessions, and would improve the information transfer between different interlocutors and local dwellers, that are, although diverse, not so different in the end. Finally, further studies should use representative sampling, in order to permit drawing wide-ranging conclusions about the whole general population from the regions.

### 4.5 Conclusions

Hazards occur regardless of the situation, whereas vulnerability is a dynamic process, always relative to a certain situation. Local knowledge, practices and cultural particularities between countries should be acknowledged, respecting local scale for every unique case and questioning risk perception. The theory of common sense helps to clarify the importance of local knowledge, keeping it from oblivion. The results from two Mediterranean case studies, North Morocco and West Sardinia,



confirm the importance of interdisciplinarity in risk management, since age, education, risk awareness sessions and hazard
history, contribute to increased precautionary behaviour of the local population. Those two case studies showed the importance of risk awareness sessions in future financial investment to protect. This information should be useful for policymakers at regional and national levels. Unfortunately, the policy literature and interviews held with the administrators and scientists indicate that although recognised, the importance of risk awareness sessions is not necessarily put into practice. As a consequence, this could lead to a failure of risk management policy. Surely, awareness-raising only is not sufficient to make
people act and take measures, but is a path that is being followed in risk management policy. Although the policy is constantly improving, with the Sendai Framework and the increase of participatory approach and local perception studies, the question is how to make a step forward. The communication gap needs to be addressed, encouraging more dialogue between different stakeholders included in the risk management process. The dire need for continuous transdisciplinary work to overcome the communication gap between the scientific community, risk administrators, and the general population is promoted. The
technical documents can be well done, but the information should be transmitted clearly, popularized, even vulgarized, to the general population, from door to door in risk areas, if needed. Last but not least, risk awareness sessions have a crucial position in learning processes and in the usage of dynamic indicators.

**Acknowledgements**

These research activities have received funding from the European Union's Horizon 2020 research and innovation programme
under the Marie Skłodowska-Curie grant agreement No713750. Also, they have been carried out with the financial support of the Regional Council of Provence- Alpes- Côte d'Azur and with the financial support of the A*MIDEX (n◦ ANR- 11- IDEX-0001-02). This work has also been funded by the RiskMED project, Labex OT-Med (ANR-11-LABE-0061) supported by the Investissements d'Avenir, French Government project of the French Agence National pour la Recherche (ANR) through the A*MIDEX project (ANR-1-1E-0001-02). The authors are grateful to the local population and other local stakeholders that
provided useful information for this study.

**Data availability**

The questionnaires and verbatim records of interviews are available, as part of PhD thesis of the first author.

**Author contribution**

The research presented in the current publication was designed and coordinated by AI, HM, and LS. The authors AI, RB, and VS designed the questionnaire and processed the data set. KA, IER, NM, MK, and OB provided expertise for parts of this interdisciplinary work and contributed to discussion.

**Competing interests**

The authors declare having no conflict of interest.



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

**Figure 1: Study sites: North Morocco and West Sardinia, Italy. Triangles point to the municipalities that were sampled.**




**Figure 2: The region of North Morocco has a high increase of touristic infrastructure.**



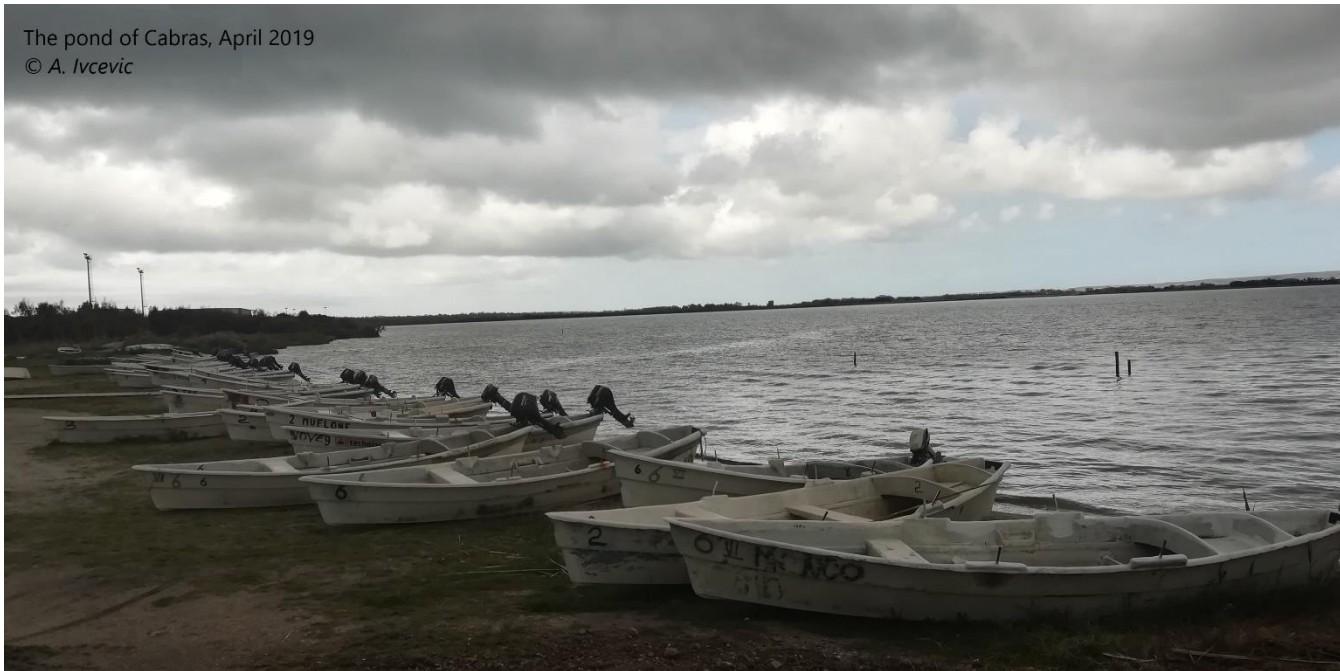

**Figure 3: Wetlands of West Sardinia are famous for its fishing activities.**






**Figure 4: Differences of main hazards perceived in North Morocco and West Sardinia by the general population**





Table 1: Variables used in regression model

| Variable | Category | N obs. | Mean | St. Dev. | Min | Max |
|---|---|---|---|---|---|---|
| Gender (female = 1) | Dummy | 567 | .4762 | .49987 | 0 | 1 |
| Age > 36 | Dummy | 565 | .5894 | .49238 | 0 | 1 |
| Education | Dummy | 539 | .3581 | .47988 | 0 | 1 |
| Zone Rif | Dummy | 567 | .2328 | .42299 | 0 | 1 |
| Zone Atlantic | Dummy | 567 | .1464 | .3538 | 0 | 1 |
| Zone Med Martil | Dummy | 567 | .3104 | .46307 | 0 | 1 |
| Urban Sardinia | Dummy | 567 | .1411 | .34843 | 0 | 1 |
| Rural Sardinia | Dummy | 567 | .1693 | .37536 | 0 | 1 |
| Personal Experience | Dummy | 566 | .5442 | .49849 | 0 | 1 |
| Precautions Taken | Dummy | 558 | .1828 | .38685 | 0 | 1 |
| Risk Informed | Dummy | 508 | .4606 | .49894 | 0 | 1 |
| Risk Sessions | Dummy | 516 | .4438 | .49731 | 0 | 1 |
| Clim. Change Belief | Dummy | 562 | .9146 | .27974 | 0 | 1 |
| Trust Science | Dummy | 555 | .5802 | .49397 | 0 | 1 |
| Trust State | Dummy | 555 | .227 | .41929 | 0 | 1 |
| Environm. Identity | Dummy | 559 | .4758 | .49986 | 0 | 1 |
| Place Attachment | Dummy | 556 | .4029 | .49092 | 0 | 1 |
| Invest Money | Dummy | 556 | .5827 | .49355 | 0 | 1 |

610          Table 2: Correlations between five different zones of the study and constructed indicators.

| Zone | Clim.ch. belief | Personal experience | Risk informed | Risk sessions | Trust science | Trust state | Place attachment | Environm. identity | Precautions taken | Invest money |
|---|---|---|---|---|---|---|---|---|---|---|
| Atlantic | .000 | -.067 | .055 | -.022 | -.002 | -.107* | .000 | -.042 | -.090* | -.024 |
| Martil | -.056 | .209** | -.064 | .009 | .145** | -.199** | -.048 | -.176** | -.018 | -.097* |
| Rif | .048 | .127** | -.051 | .220** | .022 | -.149** | -.128** | .073 | .031 | .054 |
| Urb. Sard. | .069 | -.168** | .072 | -.087* | -.208** | .365** | .038 | .014 | .045 | .037 |
| Rur. Sard. | -.049 | -.182** | .017 | -.146** | -.014 | .186** | .172** | .160** | .030 | .047 |

** significance < 0.01          * significance < 0.05






Table 3: Binary Logistic model of future financial precautionary behaviour ('Invest Money', N = 447)

| Block | Predictor | -2 Log-likelihood | Nagel kerke R² | χ² | Df | Sig | B | SE | Exp(B) | 95% C.I. for Exp(B) | |
|---|---|---|---|---|---|---|---|---|---|---|---|
| | | | | | | | | | | Lower | Upper |
| 1 | | 581.371 | .058 | 19.647 | 7 | .006 | | | | | |
| | Gender (female = 1) | | | | | | .050 | .202 | 1.052 | .708 | 1.562 |
| | Age | | | | | | -.506* | .210 | .603 | .399 | .910 |
| | Education | | | | | | .652** | .212 | 1.920 | 1.266 | 2.910 |
| | Zone Rif | | | | | | .245 | .338 | 1.277 | .659 | 2.478 |
| | Zone Med Martil | | | | | | .146 | .317 | 1.157 | .622 | 2.153 |
| | Urban Sardinia | | | | | | .486 | .387 | 1.626 | .762 | 3.471 |
| | Rural Sardinia | | | | | | .640 | .367 | 1.897 | .923 | 3.896 |
| 2 | | 566.297 | .101 | 34.720 | 11 | .000 | | | | | |
| | Gender | | | | | | .122 | .207 | 1.130 | .752 | 1.696 |
| | Age | | | | | | -.447* | .215 | .640 | .420 | .974 |
| | Education | | | | | | .528* | .219 | 1.696 | 1.104 | 2.605 |
| | Zone Rif | | | | | | .082 | .349 | 1.085 | .547 | 2.151 |
| | Zone Med Martil | | | | | | -.006 | .329 | .994 | .521 | 1.895 |
| | Urban Sardinia | | | | | | .521 | .394 | 1.683 | .777 | 3.646 |
| | Rural Sardinia | | | | | | .705 | .374 | 2.024 | .972 | 4.215 |
| | Personal Experience | | | | | | .397 | .219 | 1.487 | .969 | 2.284 |
| | Precautions Taken | | | | | | .351 | .272 | 1.420 | .834 | 2.419 |
| | Risk Informed | | | | | | .170 | .207 | 1.185 | .790 | 1.777 |
| | Risk Sessions | | | | | | .555** | .212 | 1.743 | 1.150 | 2.642 |
| 3 | | 561.319 | .115 | 39.699 | 16 | .001 | | | | | |
| | Gender | | | | | | .130 | .210 | 1.139 | .754 | 1.720 |
| | Age | | | | | | -.433* | .219 | .649 | .422 | .997 |
| | Education | | | | | | .459* | .226 | 1.583 | 1.016 | 2.465 |
| | Zone Rif | | | | | | .068 | .351 | 1.070 | .538 | 2.130 |
| | Zone Med Martil | | | | | | .008 | .333 | 1.008 | .525 | 1.935 |
| | Urban Sardinia | | | | | | .780 | .429 | 2.181 | .940 | 5.058 |
| | Rural Sardinia | | | | | | .743 | .390 | 2.102 | .980 | 4.512 |
| | Personal Experience | | | | | | .396 | .220 | 1.486 | .965 | 2.290 |
| | Precautions Taken | | | | | | .374 | .275 | 1.454 | .848 | 2.494 |
| | Risk Informed | | | | | | .118 | .212 | 1.125 | .743 | 1.704 |
| | Risk Sessions | | | | | | .539* | .217 | 1.714 | 1.121 | 2.621 |
| | Clim. Change Belief | | | | | | .181 | .496 | 1.199 | .454 | 3.168 |
| | Trust Science | | | | | | .147 | .222 | 1.159 | .750 | 1.789 |
| | Trust State | | | | | | -.410 | .270 | .664 | .391 | 1.127 |
| | Environm. Identity | | | | | | .347 | .219 | 1.415 | .921 | 2.173 |
| | Place Attachment | | | | | | .017 | .223 | 1.017 | .657 | 1.576 |

** significance < 0.01          * significance < 0.05



620          Table 4: Some examples and ideas of constructing dynamic indicators (adapted from Ivčević, 2020c)

| Discipline | Hazard Indicators | Vulnerability (pre-disaster) | Impact (response, during disaster) | Resilience (recovery, post-disaster) |
|---|---|---|---|---|
| Climatology | flooding (frequency, maximum intensity, total rainfall, floodplain inundation) | number of people living in an area at risk (basic indicator of flood exposure), land use and cover | number of households affected | number of people moved out from the area after an educational campaign |
| Climate change | relative sea-level rise, storm waves effects, significant wave height | beach nourishment and control structures | coastal storm effect and damage on shoreline | nature-based solutions to mitigate the impact of climate change |
| Geomorphology | coastal erosion rate, sediment budget, coastal slope, distance from the sea | land use and cover as indicator of financial damage | marine erosion, economic valuation of land loss | added value of tourism and of new infrastructure, ecological restauration |
| Psychology-Sociology | hazard history, last disaster date, recurrence time of earthquake, return interval of floods | risk experience, risk awareness and perception | societal response, risk awareness sessions | changes in attitudes towards precautionary behaviour |
| Religion | hazard history | information on big earthquake and flood events | risk awareness discussions in the religious institutions | adopting precautionary behaviour in order to protect lives as ethical maxim |
| Economics-Policy | disaster probability, any | cost-benefit analysis of the actual policy, proportional tax and insurance for more exposed houses | investment or insurance | cost-benefit analysis of the new policy |
| Rupture of information | Any | individual positions of different risk management stakeholders identified | number of participants and stakeholders in focus group workshops | consensual priorities of all risk management stakeholders identified |
| Management | any | number of high-school pupils educated on risk, environment and climate change | number of affected pupils | number of aware pupils, developed good understanding of climate change processes |