# Peer review of "Lessons learned about the importance of raising risk awareness in the Mediterranean region (North Morocco and West Sardinia, Italy)"

_Natural Hazards and Earth System Sciences, 2021_

## Author Comment (AC1)

Answers to Comment on nhess-2021-159

Anonymous Referee #1

The paper "The importance of raising risk awareness: lessons learned about risk awareness sessions from the Mediterranean region (North Morocco and West Sardinia, Italy)" presents several problems:

Basically, it does not respect the normal way in which a scientific paper for NHESS is prepared.

The Introduction does not report the usual review of similar papers published in literature about the topic of the paper or related topics. The papers quoted are about Mediterranean Sea. It is possible that no papers exist on similar topic? >>

Thank you for this comment. The Introduction is now rewritten. The first paragraph now refers to risk awareness: "Different societies living in one area sometimes face the same natural hazards, but do not necessarily have the same level of local risk awareness based on knowledge uptake. They do not share a common conceptual understanding of that risk, which is reflected by individual risk perception (Slovic et al., 1976). The importance of local risk knowledge and perception for sustainable disaster risk reduction and management is largely recognised by international policy, such as the Sendai Framework for Disaster Risk Reduction (UN, 2015). The question of values (Zinn, 2009), cultural and socio-economic background (Joffe, 1999; Joffe et al., 2013) are key societal problems when dealing with risks. Risk awareness and perception result in a certain precautionary behaviour which shows their relevance for risk management. However, the extent to which the local knowledge is translated into concrete, effective and practical measures is still open to debate."

Generally, at the end of the introduction, the aim of the paper is presented in a plain and simple way. In this paper, the end of the introduction reports considerations of the authors and does not explains the structure of the paper. >>

We believe that this issue comes from using sub-sections in our introduction. It gives the feeling that the 'introduction' is only the first page, not including sections 1.1 – 1.3. Because of this and for more clarity, the sub-sections will be removed and parts 1.1 and 1.3 will be integrated as part of the main introduction in the new version of the paper. The section 1.2 (study area) has been moved further in the article, under the Methodology.
In addition, the aim of the paper is presented in a plain and simple way since the study is interdisciplinary so not all the authors can enter more profoundly in theoretical psychological processes, and the targeted audience is involved in risk management, where risk research is again an interdisciplinary and a broad concept. The introduction is now enriched with the research activities: "The fieldwork procedure was both quantitative and qualitative, organized around questionnaires and interviews. In the quantitative part of the study, the used regression model was binary logistic which allows dividing variables into more blocks, tracking the model's progress, with the dependent variable of investing money. Secondly, the use of interviews in the North Moroccan study served as an additional source of information to be crossed with the responses from the questionnaires."

Concerning 1.1 Risk awareness, this section seems not effective to explain the concept directly. Moreover, the literature is quoted in a verbose way that can be rearranged to be more direct, allowing the reader to understand without reading more than once. The literature quoted is very old. The reader could think that since the past century no other papers have been published on these topics. The section is very discursive and it seems that each risk is quoted in term of example. I suggest a table for each study area, listing clearly the kind of risk and references associated to prove that it's not simply an opinion. >>

We agree that the framing of this section was not the best in term of the beginning and the end of this section, and it is now rewritten: "A leading role in efficient risk management is played by risk perception, a concept that is already widely recognized in risk literature, but still with some actual open questions. In one of the first contributions of cognitive psychology to decision making under risk, the concept of risk perception depicted that non-expert people might perceive risks by resorting to different logics than experts do (Slovic et al., 1976), making the process of risk appropriation more complex." […] "This shows that risk awareness is essential for adequate risk management policies and needs to be further explored. Moreover, since risk

questions are always societal problems and relate to values (Zinn, 2009), the cultural context needs to be tested. Inspired by a cultural approach that considers risk as a social and cultural construct (Bertoldo, 2021), this study will compare two different societies, with different socio-economic contexts. Cross-cultural differences between North Morocco and Sardinia will be detected, and the actual risk behaviour of North Moroccans and Sardinians will be described, in line with Rohrmann (1998). The focus will be more on the indicator of risk awareness and its impact on precautionary behaviour, in order to understand the practical drives for measure-taking. in addition, since it was shown long ago that the communication and the transfer of information contribute to different risk perceptions between the experts and non-experts (Slovic et al.,1976, Covello, 1983, Renn, 1998), the efficiency of risk communication between the general population, scientists and managers will be examined."

The problem with the concept of risk awareness is that it is closely related to the concept of risk perception. For years the concepts were considered as one, taking perception and awareness almost as synonyms. Even in recent risk research some authors do not seem to strictly distinguish between risk perception and risk awareness in their research (AlQahtany and Abubakar, 2020), others consider awareness to be just one of the components to measure risk perception (Khan, Rana, Nawaz, 2020). This is the reason we adopted the way of writing this section in rather chronological way.

Regardless of being a highly publishing research field in social sciences, we chose to refer to only the initial works which proposed the basic concepts we work with, and finish with the recent and the most relevant ones (by Papagiannaki et al. (2019a, 2019b)) to frame our research. We did not consider all the literature regarding examples of risks in Sardinia and in North Morocco, and we did not quote examples of risks in the area. In the rewritten version of the paper we moved the section 1.2 The study sites under the methodology (new 2.1), since it takes over the visibility from the introductory reasoning on risk awareness.

In the section 1.2 The study sites. "Being the region prone to many natural hazards and a climate change 'hotspot', the concern grows with the increase of population in already densely populated Mediterranean basin." The sentence is unclear. >>

Thank you, hopefully now is clear. The beginning of this section (new 2.1) is now rewritten: "The chosen field cases of this study are North Morocco and Sardinia in the Mediterranean basin (Fig. 1). The Mediterranean basin is densely populated region and also the most visited region in the world, with more than 330 million visitors in 2016 (Tovar-Sánchez et al., 2019). This region is prone to many natural hazards…"

**1.3 Summary and goals**

129-133 these sentences are inappropriate under the title summary and goals >>

Well noted, removed from the new version.

135-141 the meaning of the sentences is unclear. >>

This part is now rewritten: "This comparative quantitative and qualitative study contributes to explaining the role of risk awareness sessions on the preparedness to face risks. It describes relations between risk awareness and precautionary behaviour in two areas from the Mediterranean basin, North Morocco and West Sardinia. It also explores the role of risk awareness sessions in explaining the precautionary behaviour of investing money to protect. Finally, it discusses the additional benefits that risk awareness sessions can have when constructing dynamic indicators of risk management."

**2. Methodology**

The title is Methodology but actually the section contains spare sentences on the two study areas. Honestly, If I would like to apply the same "methodology" in another study area, I would have no instructions…! I suggest to use a flow chart to explain the steps of the methodology that is currently unclear. >>

The Methodology is reorganised in the new version, and the flow chart is provided. The study sites are now the initial subsection, but before it, a short paragraph is added, in order to answer to this remark: "The study was designed based on a selection of two regions from the Mediterranean basin that face some natural hazards. Then, a risk perception survey was organised among a sample of the general population using the same

main topics, but adapting to the regions' particularities. In addition, an interview guide was elaborated, and a series of interviews were conducted individually with the main stakeholders included in risk management in Morocco, to learn some lessons about the importance of raising risk awareness. A review of official legislation and secondary literature was analysed for Sardinia."

Similarly, a table of variables and their meaning could help the reader to understand something more. >>

Well noted. Table 1 of variables is now enlarged, clearly stating the questions used that lead to variables used in the study.

Moreover, the Authors should follow a clear path, describing the results of the two study areas and then comparing them, in a neat way, without all the reported examples and using tables instead of verbose description because it's difficult for the reader to understand. >>

Thank you for this suggestion. The Results and Discussion are now rewritten, with tables, in order to avoid all the quotes directly in the main text, and to facilitate the reading.

I also suggest the revision of the text in terms of language and structure. I've found several sentences that needed to be read more than once to be understood. I'm conscious that to follow these suggestions requires supplementary work but I think this could improve the paper substantially. >>

Thank you for your review and your fruitful remarks. Hopefully you would agree with our changes and answers, and approve the rewritten version of our paper.

---

## Author Comment (AC2)

General remarks

The submitted article 'The importance of raising risk awareness: lessons learned about risk awareness sessions from the Mediterranean region (North Morocco and West Sardinia, Italy)' is an interesting study about risk awareness and preparedness in 2 Mediterranean areas of different background. The subject is suitable for NHESS journal, as it addresses significant issues associated with natural hazards' risk management and adaptation. There are however several issues concerning the structure of the article, the methodology and results presentation, which need to be improved and completed for the article to be accepted for publication. A language review would also be helpful for the overall quality of the article. >>

Thank you for your review and comments that truly improve the manuscript. We hope you will agree with our proposed modifications.

**Specific comments**

In what concerns the first contact with the study objectives, the reader expects to learn a lot about risk awareness sessions, as the title implies. However, the article does not present sufficient information about these sessions, except for their impact on the willingness of the surveyed people to prepare against natural hazards' risks. This is a bit disappointing, since it would be useful for the analysis and the assessment of outputs to learn eg what kind of sessions were launched in the 2 study areas, under which framework, what was their purpose, when did they take place, were they open to the public? etc…Moreover, following these concerns, is the Title correct when it refers to lessons learned ABOUT awareness sessions? >>

Indeed, a study focused on the content of risk awareness sessions, their outputs and impact on the participants would be a useful and significant contribution to improved risk management. That would also fit better to the initial version of the paper's title. However, in this study, the focus was only on the impact of risk awareness sessions that respondents experienced personally, as a mitigation tool to prepare against natural hazards. We analysed the impact of 'risk awareness actions' in general, without a priori knowledge or focus on specific types of actions that have taken place in the study area. We agree that the title needs to be modified accordingly, and we propose it now as: "Lessons learned about the importance of raising risk awareness in the Mediterranean region (North Morocco and West Sardinia, Italy)".

**Abstract:**

L20: 'Local authorities and managers use different indicators': which ones? Please name some of them for clarity.>>

Your comment has been integrated in the abstract, it now reads: "Local authorities and managers use different indicators, usually modelled using statistical data such as % of vulnerable groups, minorities, per capita income, to quantify rescue and urbanism plans. However, existing risk indicators are often unidimensional and have certain limitations."

**Introduction**

In Introduction, the conceptual framework of the study, in which the statistical models are going to be based on, needs to be presented in a concise way. >>

In the new version the procedure is now concisely presented in Introduction: 'The fieldwork procedure was both quantitative and qualitative, organized around questionnaires and interviews. In quantitative part of the study, the used regression model was binary logistic which allows dividing variables into more blocks, tracking the model's progress, with dependent variable of investing money. Secondly, the use of interviews in the North Moroccan study served as an additional source of information to be crossed with the responses from the questionnaires.' Also, the Introduction was reorganised, removing the sub-sections and integrating them into the introduction as such, in a more concise way.

L.39: 'that risk': which one? This sounds too general, even though the article is definitive. >>

True indeed, no article needed since the discourse is on risk as a general concept. 'They might not share a common understanding of risk.'

L54. The phrase 'in order to explore what could contribute to increased precautionary behaviour in a form of future financial readiness to protect of the local, general population' is not well understood. What future financial readiness means should be clarified at this point. >>

Modified and simplified: "In this paper, North Morocco and West Sardinia, two case studies from the Mediterranean basin, are compared, in order to explore what could contribute to increased risk awareness and precautionary behaviour of the local, general population."

Sub section 1.2 would rather be a Methodology sub section. Usually the study area is described in Methodology.>>

Well noted. This section is now moved to the Methodology section.

**Methodology**

In Methodology, there should be a sub-section about statistical analyses implemented, description of the measures/concepts included in the models, etc. In general, statistical methods must be presented in a separate paragraph, following the appropriate scientific reporting: what kind of methods, and for which purpose each one will be applied; the level of significance for accepting correlation/regression results; measures should be clearly presented, how they were constructed, based on which survey questions, the items used to construct them, if multi-item assessment was applied, such as factor analyses and cronbach's alpha for scaling reliability etc. It must be clearly described which is the dependent variables, which are the independent ones (when regression is applied), which are the control variables (the dummy ones as well). >>

Well noted, in the new version an additional paragraph was added under the section 2.3 Quantitative part of the study: Survey content, variables and analysis, as well as an updated Table 1 present the precise questions used for each variable. 'The first set of variables used in the regression model consists of demographic ones: gender (value 1 for women), age (value 1 for those respondents older than the mean age of 36 years old), education (value 1 for those with bachelor's degree or higher), and zone of living (Morocco: Atlantic, Martil, Rif, Sardinia: urban and rural, five dummies for five zones). The second block includes variables related to risk awareness that are considered to describe better future willingness to invest money, as an indicator of precautionary behaviour. Risk awareness was described using questions related to the level of information about natural hazards, i.e. how much the respondents considered themselves informed about natural hazards in their region (each hazard listed and evaluated on the Likert 1-5 scale), rating their level of information about natural hazards (drought, flood, landslide, coastal storm, sea-level rise, heatwave, earthquake, coastal erosion and wildfire), and about climate change, and whether they participated or not in the awareness sessions or environmental education campaigns organized by the municipalities or associations), (e.g., Domingues et al., 2018; Ivčević et al., 2020a). The first study was the North Moroccan when the variables were explored and constructed. Items were submitted to a factor analysis (Kaiser-Meyer-Olkin Measure of Sampling Adequacy = .897; Bartlett's test of sphericity: $\chi^2$ = 1679, df = 55, p < .001), and the reliability was tested. One factor retained was 'risk informed' ($\alpha$ = .890), and the other 'risk sessions' ($\alpha$ = .381), used in the following analysis. Besides, the dichotomous variable of personal experience of a natural hazard (yes/no) was entered as dummy variable, whose importance in risk perception was showed in similar studies (e.g., Papagiannaki et al., 2019b). Moreover, respondent's belief in the existence of climate change was assessed (yes/no). Furthermore, the respondents were asked if they have previously taken any precautionary measures (yes/no). The final block of variables, related to trust, environmental identity and place attachment, was to test the supplementary variables about the impact on future precautionary behaviour. The place attachment was measured using items such as 'I think this municipality offers a good living environment' and represents a relationship of the resident with her municipality (Bonaiuto et al., 2016). These items were averaged to a single variable with a satisfactory internal

consistency (α = .827). The environmental identity was measured using items such as 'I think of myself as someone engaged in environmental issues' and represents environmental concerns of residents (Bertoldo and Castro, 2016), averaged into a single variable with a good consistency (α = .846). The social trust, regarding science and institutions involved in risk management, is of importance especially when respondents cannot manage risks on their own (Achterberg et al., 2017). The items were submitted to a factor analysis (Kaiser-Meyer-Olkin Measure of Sampling Adequacy = .768; Bartlett's test of sphericity: $\chi^2$ = 1230, df = 21, p < .001) which split them in two dimensions: trust in science (α = .887) and trust in state (α = .804). When the variables were constructed, the Sardinian data was added later when it was collected. Later, all those items were measured using a 5-point Likert scale, and then dichotomized: participants with responses above mean were considered to have a high value, those under mean to have low value, and are as such used in a binary logistic model, schematically presented in Fig. 4. For more details on variables chosen refer to publications by Ivčević et al. (2020a, 2020b).'

The questionnaire could be added as supplementary material instead of just being available upon request. This would help the reader to better understand the conceptual framework and the variables. >>

We agree. The questionnaires are now available as supplementary material.

For example, the authors mention: 'Risk awareness was described with two indicators: one related to the level of information about natural hazards (each hazard listed and evaluated on the Likert 1-5 scale), other related to risk sessions (whether the respondents participated or not in the awareness sessions or environmental education campaigns organized by the municipalities or associations)', but the reader cannot understand exactly the questions posed, for which specific hazards, what kind of information was asked to be rated? An appendix table e.g. showing the questions, the expected/constructed variables would be helpful. Furthermore, in what concerns the sessions, the only questions were about participation or were there any other questions about the content of these sessions, when they took place, and who organized them? >>

Thank you, we agree with your comment. In the new version the Table 1 was enlarged showing the exact questions posed to get the constructed variables used in the model. In addition, regarding the specification on the content of risk sessions, organized when and by whom, this was not the scope of this study and the text is now added in methodology 'It is to note that the authors of this study did not organise any risk awareness sessions in the studied region, only the previous experience of respondents regarding risk sessions was registered, as well as the information on occurrence, on who organised it and on which topic.' And additionally, under the section 4.3. Future perspectives to clarify: 'Finally, organizing a series of risk awareness sessions in the local community, opened and destined to all stakeholders, would certainly represent an instructive research activity that would contribute to clarifying the existing learning processes between the local stakeholders.'

A graphical presentation of the model would be attractive and helpful to realize the explanatory variables studied.>>

Thank you for this suggestion, a graphical presentation is now included in the manuscript (Figure 4).

Methodology presented in 2.3 subsection is not a well-understood procedure. Concepts such as indicators, risk memory, risk experience, need to be developed here, in the methodology. A table presenting the basic questions, and the interviewed groups would be helpful. As well as what method is applied to assess the answers. Also, I did not understand whether the awareness sessions asked to be rated by the surveyed population were the same discussed during the interviews. >>

Thank you for this important suggestion. The Table is added and the section is rewritten: 2.4 Qualitative part of the study: Interviews and secondary literature

In North Morocco, the interviews were based on an interview guide, which consisted of five main parts. The general topic was the influence of the representation that associates, administrators and scientists have on the relationship between citizens and risk; and on the influence of memory and the territorial context in the development of management tools, particularly preventive communication. Firstly, the life in the region and the

relationship and attachment the respondent has with the place of living was discussed. Then, the knowledge and history on risk phenomenon in the region were questioned, followed by the personal experience and risk memory. There the interlocutors were asked about the different types of knowledge (scientific and vernacular) that the institutional practices base their risk awareness activities on when talking about risks to the general population. Thirdly, the information, prevention and risk management in the region, as well as the importance of risk awareness sessions in risk management strategies were debated, followed by the consideration of the region within the context of national and international risk policies. Finally, the meaning and use of indicators in risk management were argued. There were twenty-five interviews conducted, and the interviewees were grouped into three: a) the elected or appointed administrators (eight), b) the scientists and technicians that work in the risk research or management (eight), and c) the members of civil society, i.e. associations (nine). All twenty-five narratives were recorded and transcribed and basic discourse analysis was carried out using the software Iramuteq (Ivčević, 2020c). The main parts of the interview guide and the groups interviewed are presented in Table 2. As far as the Sardinian case study is concerned, the secondary literature on risk management at the regional (Sardinia) and national (Italy) level was obtained as public documents free from the administrative web pages. These sources served to contrast the population's responses with the official directions in managing natural risks. Since this literature was easily accessible, the qualitative study in form of interviews with the local authorities was not considered to be essential for this research, conversely to the Moroccan case study.

**Results**

The use of the word indicator for the parameters/variables/measures that are correlated or regressed is not an appropriate one. Indicators of what? Please consider rewording. Moreover, if a reference on indicators of awareness or preparedness is made, it would rather be part of the discussion, which excuses a less formal writing.>>

Reworded in variables: 'Table 2 presents the correlations of five different areas (binary variable for each area) of the study with main variables.'

Table 1 does not explain properly the model variables. Which is the dependent, which are the explanatory variables, the control variables…I suggest results and tables to be reviewed by someone with good statistical knowledge. >>

Table 1 was now enlarged with additional information on variables used in the models, and questions posed in the questionnaire on which the variables were constructed.

Table 2 is not clear. Areas cannot be correlated with concepts. Future willingness to invest money IS correlated with risk awareness AMONG the people of a specific area. Is this what the authors mean? it must be clearly mentioned in the table. If the binary variable mentioned in the text corresponds to whether the respondent was or not citizen of the specific area, then what exactly is shown? That being from this area is associated with higher eg trust in science compared to the ALL the other areas? >>

We agree with your comment. A more adapted statistic would be mean comparison (one-way ANOVA): what are the means of the measures in the different areas, and how significant is their difference, and between which groups. Even more precise comparison would be Pearson's chi-square test, adapted for dichotomous variables, that does not rely on the assumption of having continuous normally distributed data. However, based on the additional input in the new version of the paper, we find this correlations part not even relevant for the main discourse. We will fully remove it from the new version of the paper.

Please keep consistency in terminology. Willingness to invest, readiness to invest, invest money, financial precautionary behavior, are all describing the same measure? If yes, please consider using one of them. >>

Thank you for this remark. Readiness to invest is now changed to willingness to invest, which is the same form of precautionary behaviour of investing money.

Sub-section 3.2.: this looks like discussion, and it is chaotic. It is difficult to follow the entire sub-section because results are not formally presented. I already suggested Tables to be included in Methodology, presenting the main topics discussed in these interviews, the interviewed stakeholders/entities. Results would therefore follow up by presenting an evaluation of the importance of awareness sessions, according to the stakeholders. An assessment must be somehow shown at this point, especially highlighting the difference between the 2 areas. >>

Thank you for this comment. This sub-section is now rewritten, with new Table 4 that sums up the verbatim regarding the stakeholders' positions on risk awareness sessions.

**Discussion**

4.1 includes statistical results that should have been included only in Results. Discussion usually does not include statistics. >>

Well noted, modified.

4.2. Statements made by the interviewed stakeholders do not help the discussion. These should be mainly part of the results. Discussion needs to help the reader understand the outcomes as filtered through results, past studies and the overall authors' assessment/aspect. >>

Thank you for your suggestion, the section is rewritten, verbatim is excluded from the Discussion and remains in the additional table that follows the part 3.2 of Results.

4.3. Again, dynamic indicators are discussed in a informal way. At last, which exactly are the suggested ones? How were they selected and how exactly they relate to this study? E.g. the 'participatory approach' is an issue aroused from the survey or the interviews? Many of the proposed indicators seem to be rather a philological discussion than revealed through this study. E.g. the content of sessions is shown as significant for the level and effectiveness of the obtained risk awareness; however, we have little knowledge about the sessions analysed in the frame of this study, as I mentioned in previous comments, thus we cannot assess the past sessions. >>

Thank you for your remark regarding this section. This truly is theoretical reasoning discussed informally. The section is, as such, rewritten, in order to underline that there were no past studies underlined, but the importance of raising risk awareness was discussed in general with the interlocutors. The section is rewritten, starting and finishing with the clarification. "Another risk management recommendation of this study regards dynamic indicators in risk management, where risk awareness sessions particularly contribute. Although the content of past risk awareness sessions was not analysed within this study, risk awareness sessions seem to be considered as useful in preparedness and risk management strategy by the participants of these research activities. As such, they could be imagined as effective risk management tool and their true utility should be tested to confirm the effectiveness of risk awareness sessions in adopting precautionary behaviour."[…] "Finally, organizing a series of risk awareness sessions in the local community, opened and destined to all stakeholders, would certainly represent an instructive research activity that would contribute to clarifying the existing learning processes between the local stakeholders and improving risk management. "

---

## Referee Report (RR1)

**General remarks**

The submitted article 'Lessons learned about the importance of raising risk awareness in the Mediterranean region (North Morocco and West Sardinia, Italy)' has been considerably improved and merits to be published. The authors have done an impressive work in this revised text, and I consider the submission of the questionnaires to be a very important contribution per se. There is now clarity on the methodology followed and the statistical analysis, while the discussion flows much better, highlighting important concerns in dealing with vulnerability to natural disasters.

**Technical comments (minor):**

L83: a verb is missing somewhere

L204: is 'associates' the exact word? could the authors mean 'associations'?

L207: perhaps 'phenomena' instead of 'phenomenon'?

L210: 'the information' sounds like something is missing (access on information, the level of information provided, informing the public…?)

L257: the scientists (plural)